# The experiences of family caregivers of people with severe mental illness in the Middle East: A systematic review and meta-synthesis of qualitative data

Aisha Hamed Alyafei[1]*, Taghrid Alqunaibet[1], Hassan Mansour[2], Afia Ali[1], Jo Billings[1]

1 Division of Psychiatry, University College London, London, United Kingdom, 2 Division of Psychology and Language Sciences, University College London, London, United Kingdom

☉ These authors contributed equally to this work.
* aisha.alyafei.18@ucl.ac.uk

**Data Availability Statement:** All relevant data are within the manuscript and its Supporting Information files.

## Abstract

### Background

There is a wealth of literature exploring the experiences of family caregivers of people with severe mental illness (SMI) in western countries, however, this topic has been neglected in the Middle East, despite families being the main source of caregiving in this context. The purpose of this review was to conduct a systematic review and qualitative meta-synthesis to explore the experiences of family caregivers living in countries in the Middle East caring for a relative with severe mental illness.

### Methods

A systematic review and meta-synthesis were conducted, to comprehensively gain a thorough and detailed overview of what is known about family caregivers' experiences from published qualitative research in the Middle East geographical area from inception to May 2021. The review protocol was pre-registered with PROSPERO (Ref: CRD42020165519).

### Results

The review identified twelve qualitative studies that explored caregivers' experiences of caring for relatives with SMI in Middle East countries. Family caregivers' experiences were captured under seven overarching themes. The participants across all studies reported negative consequences of providing care, increased burden and emotional distress. Many experienced issues with family/marital relationships and stigmatizing attitudes and behaviours from their communities. Caregivers expressed the need for increased support which was perceived to have a critical role in improving family caregivers' experiences.

**Funding:** This publication was made possible by Qatar Research Leadership Program grant, ID# QRLP11-G-1902002, which is under Qatar National Research Fund (A member of Qatar Foundation). The funders had no role in study design, data collection and analysis, decision to publish, or preparation of the manuscript.

**Competing interests:** The authors have declared that no competing interests exist.

## Conclusions

The meta-synthesis revealed many challenges and issues that affect caregivers of people with SMI in the Middle East. Family caregivers experienced distress and burden, and reported significant impact on their psychological well-being. Their experiences highlight the urgent need to provide more support for family caregivers in Middle East countries.

## Introduction

Severe mental illnesses (SMI) are mental, behavioural, mental, behavioural and emotional disorders that cause substantial functional impairment that interferes with or limits daily activities and include disorders such as schizophrenia and bipolar disorder [1]. Common signs and symptoms of SMI include hallucinations, delusions, severe depressive episodes, and manic symptoms [2]. Recent studies have reported that mental illness is a major global issue, with more than 45 million people worldwide affected with bipolar disorder, and around 20 million people worldwide affected with schizophrenia [3].

Family caregivers are often an important source of care for people with SMI. They may provide practical and personal, as well as emotional and financial support for their relatives [2,4]. People with SMI often depend heavily on their carers, and the status of their illness, therefore, frequently depends on the quality of care provided by them [5]. In western countries, much research has been conducted into exploring the experiences of family caregivers of people with SMI. This research has noted that providing care constantly affects the whole family in many ways [6]. Caring for relatives who have SMI can affect the physical and psychological well-being of the caregivers, as well as their ability to cope with the situation [7]. A national survey conducted by The European Federation of Families of People with Mental Illness, was distributed to a sample of carers from different European countries and reported that caregivers of people with SMI experienced cumulative burden across multiple life domains such as emotional, social, physical, financial and relationship burden. The study found that more than one in three family caregivers felt that they could not continue their role and had reached 'breaking point' [8]. Caregiving has also been linked with negative impacts on quality of life [9,10]. As carers are often overwhelmed by the demands of caring for their relatives with SMI, they are frequently under considerable stress [8,10]. Studies have reported that the impact of caring for a relative with mental disorder can confer risks of mental ill health, such as depression, to carers themselves [11–13]. Further, according to a recent review conducted on violence by people with serious mental illness toward family caregivers, Labrum (2020), found that family caregivers often experienced violence when they are at home alone with their relative with serious mental illness, which can cause a significant negative impact on the wellbeing of caregivers [14]. Most of this large body of research has, however, so far been conducted in Western countries. In the Middle East, very little research exists regarding family caregivers of people with SMI, despite the fact that the family unit plays a fundamental role in caregiving for ill relatives in this context.

According to the Center for Middle Eastern Studies at the University of Chicago, the Middle East region include countries of Southwest Asia and Northeast Africa (Qatar, Saudi Arabia, Bahrain, United Arab Emirates, Kuwait, Oman, Yemen, Iraq, Lebanon, Palestine, Israel, Jordan, Syria, Turkey, Iran, and Egypt) [15]. Islam is the dominant religion in this region. Almost 90% of people in this region are Muslims and they consider Islam not only a religion but also a way of life [16]. The practice of the Islamic religion in this region has a great impact on people's attitudes, spiritual beliefs, and other aspects of day-to-day living [17]. In relation to

family, Middle Eastern cultures have conservative attitudes, as families are considered to be the heart of every Muslim society [18]. They are the main source of support to relatives and moral virtues and values such as love, mercy, good habits, and compassion are promoted in the family setting and passed on to future generations [18,19]. Providing care for an ill family member by a family caregiver is rooted in cultural beliefs as well as in the Islamic faith which encourage family members to ensure their relatives' wellbeing [4]. Despite the importance of family and its duty in Muslim Middle Eastern communities, emerging research shows that family caregivers may sacrifice their own needs for the benefit of their relatives [19]. In the Middle East, providing care is more likely to be a culturally appreciated practice, but also a cultural obligation [20]. Research on family caregivers' providing care to ill relatives in this context is, however, currently very limited, with the experiences of family members caring for people with SMI having been largely neglected in Middle Eastern countries [21].

There is very limited evidence available to date that addresses this topic, and where studies do exist, they have been mostly quantitative in nature; examining caregiver burden as measured by questionnaires and scales [22,23]. Few studies have explored in depth the experiences of caregivers in this region using qualitative methodologies. Some research has been conducted on the experiences of caregivers caring for relatives with different mental health needs. One study explored the experiences of caregivers of elderly people in Qatar and found that stress and burden was one of the main factors that negatively impacted the caregivers [24]. Some research has been conducted on caregivers' experiences of caring for people with Alzheimer's disease in Israel and Iran, which demonstrated that caregivers experienced stigmatizing behaviors from the public and family. The authors described that some of the perceived stereotypes provoked negative emotions in caregivers such as shame and embarrassment [25,26].

Although experiences of caregivers in the Middle East may be similar to those experienced by carers for people with SMI in western countries, it is important to acknowledge the influence of cultural and societal factors, such as religious and spiritual aetiology and mental illness stigma, which may be more prominent in the Middle East. Although mental health diagnoses are universal, the manifestation of mental disorders may vary in each culture and cultural factors shape attitudes in each population [19]. For example, in some Arab cultures, a commonly held belief is that mental illness originates from evil spirits or punishment from God [19,27]. It is crucial that we better understanding family caregivers' experiences in Middle Eastern countries in order to better support their potentially unique views and needs.

Whilst Middle Eastern countries can be distinguished from the West, it is important to acknowledge that there are also cultural, economic, religious and language variations between the different Middle Eastern countries. For example, there are differences in terms of understanding the causation of mental illness, services and funding available to treat mental illness, the degree of family involvement into caregiving, and available resources to support families. As such, it is important to look to qualitative studies to produce a more detailed understanding of caregiver experience with, and between, different Middle Eastern countries [28,29].

There is currently a dearth of research into the experiences of family caregivers in Middle Eastern countries, despite caregiving predominantly being provided in a family environment in these countries. Therefore, in this review we set out to explore and meta-synthesise qualitative literature describing caregivers' experiences of caring for a family member with SMI in Middle East countries. By providing a detailed meta-synthesis of data from across included studies, this review will help to bring together all the qualitative data in this field so far to strengthen the evidence and form a new interpretation about caregivers of people with SMI in the Middle East. The results of this review will then guide a better understanding of this issue and the development of recommendations for supporting family caregivers in this context.

## Methods

### Design

A systematic review was carried out to identify studies and systematically select data related to the experiences of family caregivers living in Middle East countries caring for a relative with SMI.

The systematic methodology associated with this meta-synthesis provides the opportunity to gain a thorough and detailed overview of what is known about family caregivers' experiences from published qualitative research in this geographical area to date. Given that research on this topic in this geographic region is still in its infancy, we decided to focus on only qualitative studies in order to provide us with much richer sources of information about caregivers' experiences.

The review protocol was pre-registered with PROSPERO (Ref: CRD42020165519). PRISMA guidance was adhered to throughout this review in order to ensure clarity and transparency of reporting and evaluating studies (see S1 File. Appendix for PRISMA checklist) [30].

### Search strategy

We conducted our final systematic searches of qualitative research studies on the 3rd of May 2021, across seven major electronic databases including CINAHL, PsycINFO, Web of Science, MEDLINE, ProQuest Family Health, Qatar National Library and QScience. We also searched grey literature through Google Scholar, OpenGray, ProQuest Dissertations & Theses Global, and EThOS to identify any additional published papers. 'Hand searching' was also conducted by screening the contents of relevant Arab journals such as *Egyptian Journal of Psychiatry* and *The Arab Journal of Psychiatry* to find suitable articles, which may not have been picked up by the databases or missed during indexing. Backwards and forwards citation checking was conducted for all included papers.

Key search terms were derived from four main search terms "family caregivers", "experiences", "mental illness", and "Middle East". We applied the search terms across sources using advanced search options, the PICO structure and MeSH terms combined using Boolean operators. For this search process, we consulted the university librarian to support the search strategy. These main search terms were further elaborated to include alternative terms relevant to each database (see S2 File. Appendix for full search terms).

### Selection criteria

Specific pre-selected inclusion and exclusion criteria were used to inform our search. The inclusion criteria were that studies had to (1) use a qualitative research design (for mixed method studies: there must have been qualitative data which could be extracted and analysed independently), (2) be published journal article, theses, or dissertations, (3) be from inception up to May 2021, (4) be from Middle East countries as defined by the Centre for Middle Eastern countries [15], (5) be in any language as long as there was an English or Arabic abstract, (6) consist of a sample of family caregivers 18 years old and above who had cared for a relative with severe mental illness for more than one year, and (7) include patients diagnosed with SMI (schizophrenia, bipolar disorder according to DSMI-V or equivalent showing signs and symptoms of psychosis like hallucinations and/or delusions).

Studies were excluded if they were (1) unpublished studies (e.g. conference abstracts, trial protocols, or studies reported in book chapters, editorials, reports, letters, or general comment papers), (2) case studies, and previous reviews, (3) quantitative studies, (4) paid carers or individuals who have never experienced caregiving or were providing care to family members with

physical illnesses or non-SMI mental illnesses (e.g. cancer, diabetes, intellectual disability, developmental disorders, autism, dementia, stress, depression etc.).

## Data screening and extraction

All search results were imported into Endnote for sorting. Duplicate articles from more than one database were removed. AHA and TA worked independently and screened all titles and abstracts of papers for relevance following the inclusion and exclusion criteria. Articles that did not meet the inclusion/exclusion criteria were removed. Full texts of the remaining articles were then screened independently by the two authors to confirm eligibility. Disagreement between the two authors at any stage was resolved through discussion over the eligibility of the studies. The extracted data from the selected studies were imported into a pre-designed data extraction table, which was developed by the first author, AHA, to organize the relevant information based on methodologies, sample size, and other related information, see Table 1. All relevant studies were included in the qualitative meta-synthesis.

## Quality appraisal

The overall quality of each study was evaluated according to the Critical Appraisal Skills Programme (CASP) qualitative appraisal checklist which is a widely used instrument for assessing the quality of qualitative study designs. The checklist contains guidance on appraising the methodological quality of all qualitative studies and a consideration of the validity and transferability of the results [31]. This process was carried out by AHA and HM independently and discrepancies were resolved through discussion with the rest of the authors. We followed guidance by Lachal et al. (2017) in using the CASP checklist and classified studies according to whether they totally met, partially met, or did not meet the quality criteria [32].

## Meta-synthesis

We followed the process described by Lachal et al. (2017) on synthesising qualitative literature in psychiatry [32]. From all included articles, data about family caregivers' experiences was extracted from results sections and entered verbatim into NVivo Pro version 12 for qualitative thematic analysis.

In line with thematic analysis, each article was read and reread carefully in order to become familiar with the data through "active reading, with the intention of appraising, familiarising, identifying, extracting, recording, organising, comparing, relating, mapping, stimulating and verifying" [32]. AHA generated the original initial codes from the reading and re-reading of all included studies which were relevant to this review question. JB and AA reviewed the initial codes that were generated by AHA and based on their feedback the initial codes were revised.

By examining similarities and differences between initial codes as well as grouping and categorising them into a hierarchical structure, we developed a set of descriptive themes. These themes were compared across articles "to match themes from one article with those from another while ensuring that each key theme captured similar themes from different articles" [32] (p. 6). We developed an overarching, original, set of themes and sub-themes that captured the experiences of family caregivers of people with SMI in the Middle East. To enhance validity, the emerging themes were discussed with all the authors.

## Reflexivity

Reflexivity is widely used in the context of qualitative research methodology. It is a key measure of quality in qualitative research [33]. Through reflexivity, reviewers and readers can

**Table 1. Characteristics of included studies.**

| Study No. | Author(s) (Year) | Study country | Sample size/population studied | Type of study | Aims | Method | Analysis |
|---|---|---|---|---|---|---|---|
| 1 | Akbari et al. (2018) [34] | Iran | n = 20 (mixed first-degree relatives, female carer: 60% and male carer: 40%) | Qualitative study | To explore the support needs of family caregivers of people living with a mental illness in Iran | Individual interviews | Content analysis |
| 2 | Alasmee et al. (2020) [35] | Jordan | n = 21 (mixed first-degree relatives, female carer: 61.9% and male carer: 38.1%) | Qualitative study | To explore family caregivers experience towards antipsychotic medications. | Semi-structured interviews | Thematic analysis |
| 3 | AlMakhamreh et al. (2017) [36] | Jordan | n = 14 (mixed first-degree relatives, female carer: 83.1% and male carer: 16.9%) | Mixed methods | To generate meaningful understanding of the mental health informal carers' experience and to identify a possible approach to social work intervention. | Focus group | Thematic analysis |
| 4 | Amsalem et al. (2018) [37] | Israel | n = 15 (mixed first-degree relatives, two couples of parents, six mothers, six fathers, and one partner) | Qualitative study | To explore the subjective experience of stigma and macroaggression among consumers and their family members during their encounters with mental health care providers. | Semi-structured interviews | Content analysis |
| 5 | Attepe Özden and Tuncay (2018) [38] | Turkey | n = 31 (mixed first-degree relatives, female carer: 35% and male carer: 39%, for siblings their gender is not specified: 26%) | Qualitative study | To understand the personal burdens that affect families that provide care for individuals with schizophrenia and what they are using as coping strategies, their social support role and their need. | In-depth interviews | Thematic analysis |
| 6 | Ebrahimi et al. (2018) [39] | Iran | n = 16 (mixed first-degree relatives, gender characteristics not specified) | Qualitative study | To explore barriers impeding family caregivers' ability to cope with their relatives diagnosed with severe mental illness | Semi-structured in-depth interviews | Content analysis |
| 7 | Rahmani et al. (2018) [40] | Iran | n = 14 (female spousal only) | Qualitative study | To explore the experiences of female spousal caregivers in the care of husbands with severe mental illness | Semi-structured in-depth interviews | Conventional content analysis |
| 8 | Sari and Duman (2020) [41] | Turkey | n = 16 (mixed first-degree relatives, female carer: 81% and male carer: 19% | Qualitative study | To reveal experiences of family caregivers of individuals with chronic psychiatric illness | Semi-structured interviews | Content analysis |
| 9 | Shamsaei et al. (2015) [5] | Iran | n = 16 (mixed first-degree relatives, female carer: 69.2% and male carer: 30.8%) | Qualitative study | To explore the challenges with which the family caregivers of patients with chronic mental illness have to contend | In-depth semi-structured interviews | Colaizzi phenomenological analysis |
| 10 | Sharif et al. (2020) [29] | Saudi Arabia | n = 13 (mixed first-degree relatives, female carer: 77% and male carer: 23%) | Qualitative study | To explore the experiences of family caregivers of people living with various mental disorders through examining the burdens that they face and the coping strategies that they use. | Semi-structured interviews | Thematic analysis |
| 11 | Tamizi et al. (2020) [28] | Iran | n = 12 (mixed first-degree relatives, female carer: 75% and male carer: 25%) | Qualitative study | To gain a better understanding of caregiving burden in family caregivers of patients with schizophrenia and its related factors | Semi-structured interviews | Content analysis |
| 12 | von Kardorff et al. (2016) [42] | Iran | n = 45 (mixed first-degree relatives, female carer: 70% and male carer: 30%) | Qualitative study | To explore the specific burdens experienced by caregivers of patients with schizophrenia and affective disorders | Semi-structured interviews | Content analysis |

understand how researchers' perspectives and assumptions may have influenced this study and the interpretation of the data included in the meta-synthesis. The reader is able to consider the validity and credibility of the study results by better understanding the research team who have produced them. It also enables the reader to acknowledge the personal experiences, interpretations and preconceptions that the researchers have brought to the study from different

professional backgrounds and helps to understand the lenses through which the researchers have understood the data. This review involved a multidisciplinary team, because such collaborative research improves the quality and rigour of the qualitative synthesis. The research team was made up of a diverse group of researchers including Middle Eastern Arabs, South Asian British and White British researchers. The research team are at different career stages from Doctoral students to senior academics with particular expertise in severe mental illness, systematic reviews and qualitative meta-synthesis.

## Results

From the electronic database search, we retrieved a total of 831 records. A further six studies were identified through hand searching of reference lists and two potential studies were identified by a subject matter expert. Duplicate publications were removed, then the titles and abstracts of 354 studies were screened for relevance by two authors. A total of 312 studies were excluded at this stage resulting in 42 studies which were then reviewed in full by the two authors. Thirty studies were excluded as they did not meet the inclusion criteria, one pilot study was removed because both the pilot and the main study used the same cohort, therefore, only the main study was included in the review, and one PhD thesis was removed due to a published version of the study being available, which was included in the review. This resulted in twelve studies which met all the inclusion criteria and were included in the systematic review and meta-synthesis. See (Fig 1) for PRISMA flowchart and study selection process.

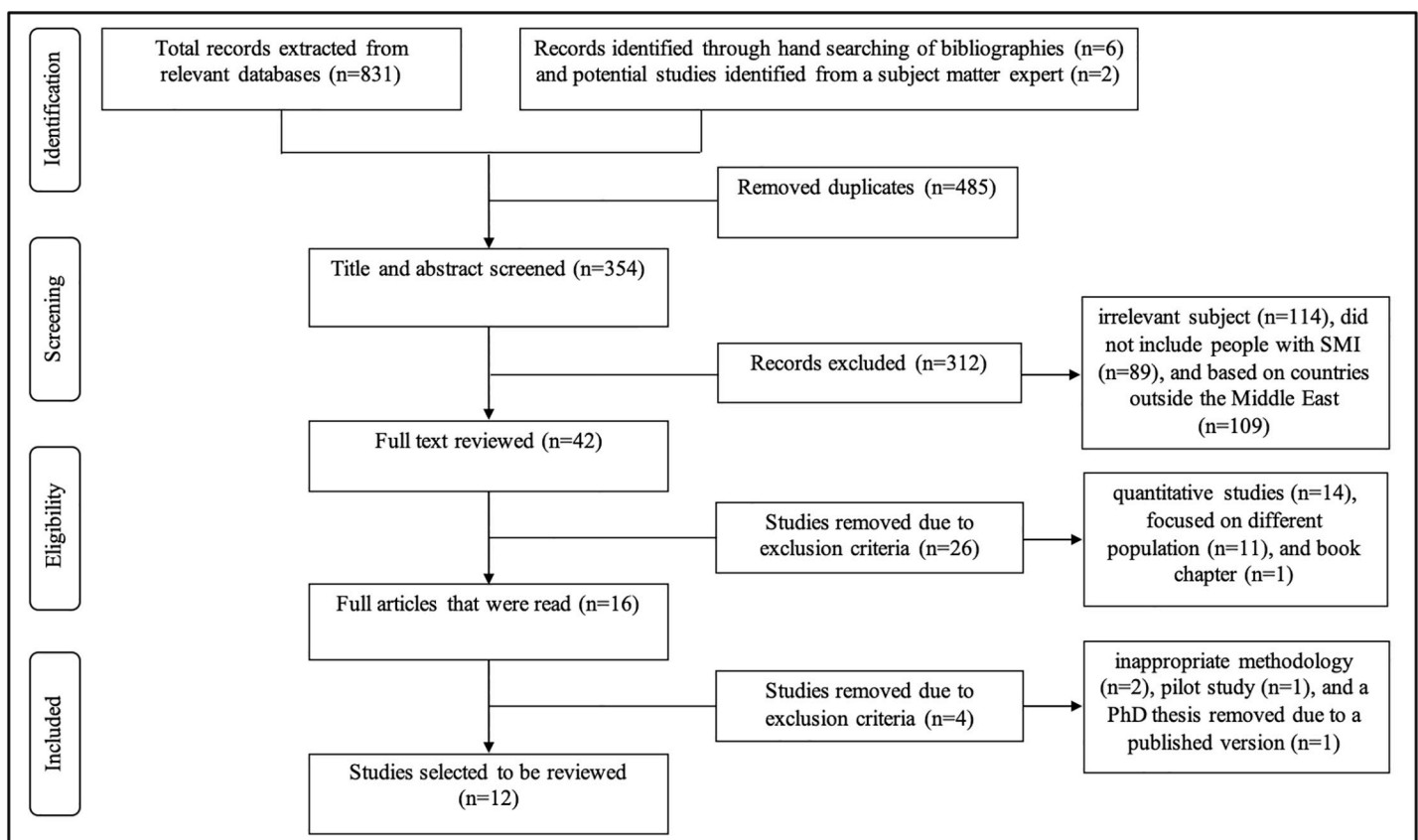

**Fig 1. PRISMA flowchart-study selection process.**

## Characteristics of the selected studies

The twelve included research studies had sample sizes ranging from 12 to 45. Six studies were undertaken in Iran, two from Turkey, two from Jordan, one from Israel, and one from Saudi Arabia. The participants from most of the studies were described as mixed first-degree family caregivers of relatives with different mental disorders such as schizophrenia or bipolar disorder. Only one study focused on female spousal caregivers of male spouses with SMI.

From different academic disciplines, all the studies aimed to explore the experiences and challenges of family caregivers of relatives with SMI. All studies were peer-reviewed and published in academic journals between 2015 and 2020. In most of the studies, data were collected through individual semi-structured interviews. Only one study collected data through focus groups. See Table 1.

## Quality appraisal

We assessed the quality of the twelve included studies using the CASP checklist. We did not exclude any studies for this review on the basis of their quality due to the lack of research related to this topic. The quality ratings for each individual study are shown in Table 2.

## Themes

From our meta-synthesis of the qualitative data seven overarching themes were generated, see Table 3. These themes are presented below with direct quotes from the included studies used to illustrate each theme. The relationships between themes are depicted in (Fig 2) which provides a visual depiction of the overarching themes and their interrelated nature.

**1. Lack of acceptance, denial and misunderstanding.** This theme relates to the caregivers' lack of acceptance, denial and misunderstanding of the diagnosis of their relatives. Family caregivers who were interviewed in the included studies shared different experiences, therefore, the theme is further divided into two sub-themes: lack of knowledge and caregivers in denial.

*1.1. Lack of knowledge.* Many family caregivers interviewed in the included studies were shocked after their loved one was diagnosed with SMI. A mother described how she was surprised and uncertain about how she should respond when her daughter was diagnosed with a mental disorder. She was not prepared for this role, she noted:

*"We were very worried at that time. I was surprised and didn't know what to do. What shall I do? How shall I treat her?"* [38] (p. 499).

Due to their lack of knowledge, some caregivers initially looked to alternative interpretations. A father initially attributed his son's illness to supernatural forces and acknowledged that it took him several years to understand the true nature of his son's illness:

*"We, the Turkish nation, have something. People think little monsters have attacked, or djinns or fairies have attacked. All of them are wrong. We've understood all of them, but do you know how I've understood them? I was able to understand it in two to three years"* [38] (p. 501).

Lack of understanding by the caregivers was compounded by patients themselves not having insight which led to difficult family dysfunction and distress. Family caregivers were often challenged in trying to manage their loved one with SMI. A caregiver shared their experience about their family member's paranoia towards them, which led to criminal proceedings:

**Table 2. Full CASP assessment on twelve studies.**

| CASP questions | Study numbers | | | | | | | | | | | |
|---|---|---|---|---|---|---|---|---|---|---|---|---|
| | 1 | 2 | 3 | 4 | 5 | 6 | 7 | 8 | 9 | 10 | 11 | 12 |
| | Akbari et al. (2018) [34] | Alasmee et al. (2020) [35] | AlMakhamreh (2018) [36] | Amsalem et al. (2018) [37] | Attepe Özden & Tuncay (2018) [38] | Ebrahimi et al. (2018) [39] | Rahmani et al. (2018) [40] | Sari & Duman (2020) [41] | Shamsaei et al. (2015) [5] | Sharif et al. (2020) [29] | Tamizi et al. (2020) [28] | von Kardorff et al. (2016) [42] |
| 1. Was there a clear statement of the aims of the research? | T | T | T | T | T | T | T | T | T | T | T | T |
| 2. Is a qualitative methodology appropriate? | T | T | T | T | T | T | T | T | T | T | T | T |
| 3. Was the research design appropriate to address the aims of the research? | T | P | N | P | T | T | T | T | T | T | N | N |
| 4. Was the recruitment strategy appropriate to the aims of the research? | T | T | P | T | P | T | T | P | T | T | P | T |
| 5. Were the data collected in a way that addressed the research issue? | T | T | N | T | T | T | T | T | P | T | P | T |
| 6. Has the relationship between researcher and participants been adequately considered? | N | P | P | T | N | N | N | P | P | P | N | N |
| 7. Have ethical issues been taken into consideration? | P | T | T | T | P | T | T | P | T | T | T | T |
| 8. Was the data analysis sufficiently rigorous? | T | T | P | T | T | T | T | T | T | T | T | T |
| 9. Is there a clear statement of findings? | T | T | T | T | T | T | T | T | T | T | T | T |
| 10. How valuable is the research? | T | T | T | T | T | T | T | T | T | T | T | T |

*T = Totally met, P = Partially met, and N = Not met.*

*"My patient has already complained to the court more than 10 times before [about me], but I have been exonerated". "There was no crime here. The patient was paranoid, and he thought that the family members were the enemy. When the judge realized this, then the caregiver was acquitted"* [34] (p. 900).

**Table 3. Overarching themes emerging from meta-synthesis.**

| Key themes | Sub-themes |
|---|---|
| 1. Lack of acceptance, denial and misunderstanding | 1.1. Lack of knowledge |
|  | 1.2. Caregivers in denial |
| 2. Burden of Caring | 2.1. Self-sacrifice and responsibility |
|  | 2.2. Physical burden |
|  | 2.3. Financial burden |
|  | 2.4. Burden of caregiving on women |
|  | 2.5. Domestic abuse and violence |
|  | 2.6. Concerns and doubts about the future |
| 3. Damaged and disrupted relationships | - |
| 4. Impact of caring on caregiver's mental health | - |
| 5. Stigma | - |
| 6. Seeking support | 6.1. Social support |
|  | 6.2. Lack of Professional support |
| 7. Attempts to cope | - |

*1.2. Caregivers in denial.* When family caregivers were looking after their relatives with SMI, they expressed difficulties accepting the disorder and coming to terms with the diagnosis. One mother expressed her experience of trying to accept the disorder, she stated:

*"We act as if we have accepted it. However, indeed, we haven't accepted it. We have questions in our minds at every moment"* [38] (p. 500).

Caregivers struggled to accept and accommodate the course and prognosis of SMI and how it could affect individuals later in their life.

*"I had difficulty in believing that a person who had graduated from university suddenly had schizophrenia. If she had had it from the beginning, it would have been easy to accept that it became worse and worse in the course of time . . . (Talking about his daughter İlknur, who was diagnosed while working after university graduation)"* [38] (p. 500).

Several family caregivers turned to religious explanations. One family caregiver from Jordan accepted the situation as God's test for them, they stated:

*"They believed they were caring for a good cause: God will reward me with my children for my patience. They will grow and help me. This is what keeps me alive. Caring is a test and destiny (nassep) from God"* [36] (p. 1047).

One mother from Saudi Arabia shared similar experience of how being close to God helped her to accept her daughter's illness:

*"All praise to God, we accept it because we know God is generous, eventually, we can't escape from fate, God will be above everything"* [29] (p. 9).

**2. Burden of caring.** The presence of a mental disorder in the family contributed significantly to an increase in caregiving burden, with many caregivers feeling that they had to take sole responsibility for many aspects of home, personal and financial life. Many family caregivers reported that they experienced negative consequences of providing care to relatives with

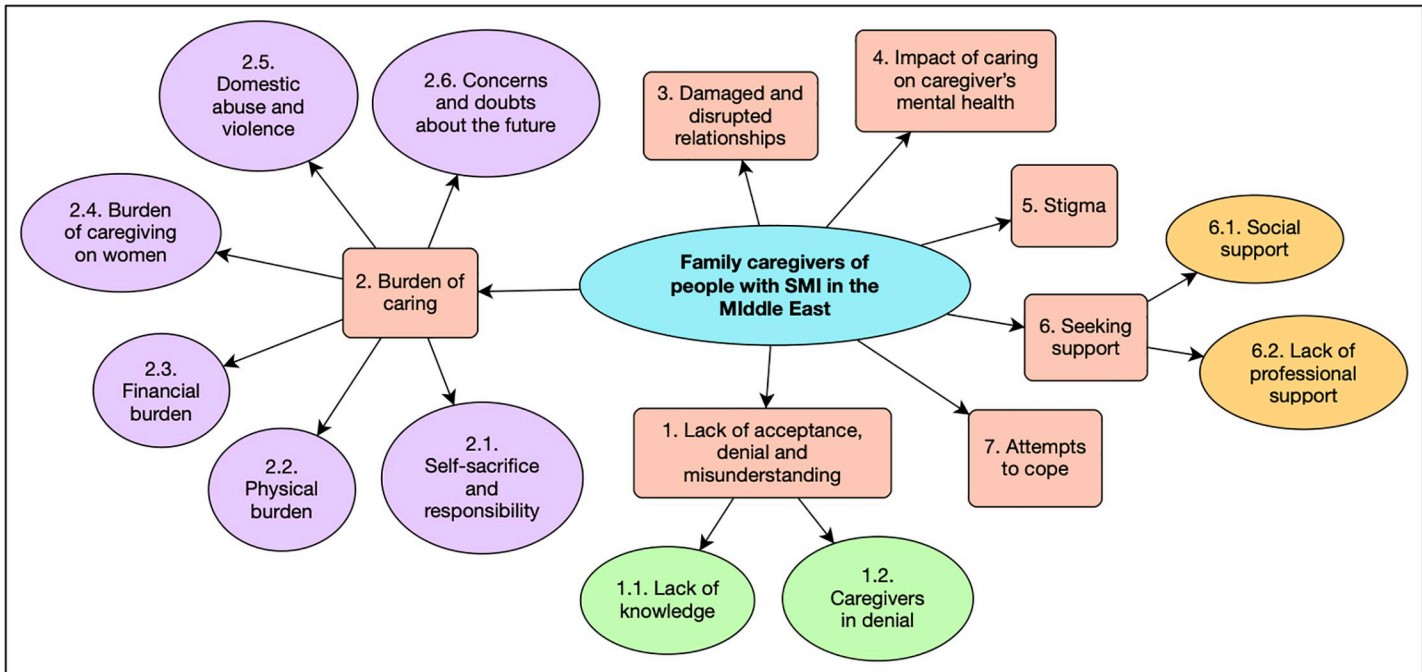

**Fig 2. Overarching themes.**

SMI. They gave up things that they previously enjoyed doing and spent most of their day monitoring their relative's mental health, and having to endure behavioural disturbances, and frequent relapses.

*2.1. Self-sacrifice and responsibility.* Struggling with balancing the demands of caregiving alongside managing daily tasks, family caregivers felt restrained and were forced to play different roles while providing care. The roles and impact of caregiving were different according to whether the caregiver was a mother, father, or a spouse. According to Rahmani et al. (2018) [40], a female spouse described herself as feeling trapped in her relationship because of her competing responsibilities and obligations:

> *"Sometimes, I perceive myself like a butterfly who has been trapped in a spider's web. I'm helpless in my life. I'm trapped between my different roles in my life. What can I do? Which one should I put in priority? I do not know"* (p. 1512).

Family caregivers were often the only source of support for people with SMI and this could involve constant supervision and monitoring. As a result, caregivers could not go about their daily life freely. A sibling caregiver described restrictions in various daily activities:

> *"I can't go anywhere. I can't go on a trip. I must always accompany her and look after her. Whenever I go out, I have to come back early"* [42] (p. 250).

Caregivers often had to help their family member with activities of daily living such as cooking and eating, giving medications, attending hospital appointments, attending to personal hygiene, and grooming. A mother from Iran declared how she did everything for her adult son who had been diagnosed with schizophrenia:

*"I have to do everything for my son. He cannot do anything on his own. I have to supervise on everything from eating to taking medicine"* [28] (p. 5).

Most of the responsibility for looking after someone with SMI fell on female spouses, who often felt distressed and overwhelmed. One female spouse expressed how life has become very exhausting because of her husband who has a mental disorder and requires full-time care. He is totally dependent on his wife and the rest of the family. He lost his ability to perform daily life activities due to his illness.

*"Despite the long years of my husband's illness, he is still unable to take care of himself. He forgot that he can do some things. He is totally dependent on us. It seems that he is accustomed to depend on someone else. I will always take care of him. Constantly. . . every moment. . . It was really bothering me. He is completely passive. No matter what's going on around him"* [39] (p. 990).

With multi-task roles, responsibilities and time spent caring for family members with SMI, most of the family caregivers gradually neglected themselves and no longer prioritised their own lives. With overwhelming demands and stress due to caring, a female spouse caregiver described herself as the "walking dead":

*"I'm like a walking dead and I don't know what I want for myself; I've forgotten my femininity. I think I'll become a person who should not think of herself at all, who should always take care of her family and never miss anything. I've forgotten myself. Sometimes, I forgot who I used to be before, what I wanted"* [40] (p. 1513).

*2.2. Physical burden.* Participants in many of the studies described the physical burden of caring for a family member with SMI, noting the impact on their physical health, leading to them feeling exhausted and tired.

Physical burden such as lacking energy or experiencing pain affected the caregivers' ability to function. One family caregiver captured the physical impact of caregiving and expressed their needs for support to share the burden:

*"I am tired and I need help, I need someone to share the burden, why mental health is not important like physical illness, I am losing energy, sometimes I cannot breathe, I feel I am going crazy"* [36] (p. 1050).

A 61-year-old sibling caregiver from Turkey who had spent 33 years providing care to a relative with mental disorder, described her experience with a very physical metaphor, as akin to a donkey carrying a heavy physical load. She reported:

*"What does it resemble in life? I mean it resembles a donkey. Yes. It will be constantly loaded but will not say anything, because it cannot say anything. The most it can do is to give a saddened look with its beautiful eyes. That's all. I mean its eyes are always sad; donkeys have the world's most beautiful eyes. But it is always sad, always destitute. Because it always carries a load"* [41] (p. 42).

*2.3. Financial burden.* Family caregivers often reported that their loved one could not work due to their mental ill health. Consequently, the family members faced a significant financial burden.

One female spouse from Iran reported that her husband's illness and his inability to return to work, created a 'disaster' for the family:

*"When he got better, he could no longer go back to his work; he was no longer hired and wasn't allowed to work there again. It was a disaster for the whole family. Moreover, he can no longer perform paternal duties or assume his responsibilities as before"* [39] (p. 991).

Caregivers felt that providing care to their relative with SMI also affected their ability to work and led to financial concerns, particularly due to the high medical costs associated with the illness:

*"Sometimes I have to work less to manage my family matters, and ultimately I have to work fewer hours than desired. In the end, I get less pay and with all this extra cost, I'm really helpless. I wish that there would be rules that support us in such cases"* [39] (p. 993).

A mother experienced financial burden because she sought treatment from multiple spiritual heelers rather than mental health professionals:

*"We've also gone to many so-called divine healers. Be sure of that, we've given a lot of money. We've borrowed money from others. We've given 500, then another, and so on. (Seeking help from divine healers rather than professionals.)"* [38] (p. 501).

*2.4. Burden of caregiving on women.* In all the included studies, caregivers were mostly women which is common practice in the Middle East countries. This is consistent with traditions and teachings about family responsibilities and relationships that most Middle Eastern communities are raised on.

Women were the main source of care and compassion in the family. According to AlMakhamreh (2018) [36] study, women caregivers from Jordan reported that they were expected to provide care as it was their social duty:

*"This is how we were raised in our house; we have to take care of all elderly people and our children; women are trained to do the caring"* (p. 1047).

*"I don't think anyone is good in caring but women, we have an old saying 'if you raise a good woman you raise a whole good tribe'"* (p. 1047).

*2.5. Domestic abuse and violence.* Caregivers were also often subject to episodes of anger and violent behaviour from their loved one with SMI, impacting on their personal safety. Caregivers had insufficient knowledge and awareness of how to manage these high-risk behaviours and often felt helpless.

A well-educated 34-year-old sister caregiver who was employed in part-time job, faced difficulties when her sibling who had schizophrenia became angry:

*"When she broke her mobile phone and laptop, I didn't know how to calm her down, how to deal with her, and I had been very afraid"* [28] (p. 5).

Another sister caregiver reported concerns about her brother's behaviour who might harm the family.

*"I really do not know what to do. I feel a lot of stress, especially at bedtime. I am afraid he might do something to harm us, so I collect all the dangerous items in the room, I also gather the fruit knife for food. I do not sleep well. I worry about it . . ."* [34] (p. 900).

A female spouse gave a harrowing account of being forced into isolation and of marital rape.

*"He always asks me where I go, what I do, sometimes isolates me for a month. Then all of sudden he asks the family to go out and then he rapes me"* [36] (1049).

Whilst family caregivers often experienced anger and violence from their loved ones, they were also often concerned about their loved one harming themselves. A caregiver from Iran felt helpless and sad due to their sibling's self-harming behaviour and suicide attempts:

*"I felt so sad because I could not bear that he hurts himself. I could not bear his attempts to commit suicide, because I could not do anything"* [42] (p. 250).

*2.6. Concerns and doubts about the future*. Caregivers expressed their concerns about the future of their loved one. They had a range of questions in mind. A father shared his doubts and concerns for the future:

*"I have felt excessive stress and sadness from all of this. It's just always on your mind–wondering what's going to happen next, what will happen when we're not around, or when we're not able to care for him"* [5] (p. 4).

**3. Damaged and disrupted relationships.** According to the experiences of caregivers, they reported having poor or limited relationships with the rest of their family members. This was due to the burden of their caring role and the impact of caring for their relative on the wider family system.

A female spouse described the impact of her husband's mental illness on her marital relationship, leading to her becoming frustrated and emotionally detached:

*"Sometimes I feel like I am a slave. Do you know slave? Every time I have to go out, I have to take his permission. He often accuses me of something I've never done. I can't argue or resist. This causes me to get angry with him. However, I can't express it to him. That pushes me away from him"* [40] (p. 1511).

A female spouse from Jordan, described her relationship with her husband as an 'unbearable' experience and she expressed her need for professional support:

*"I need advice for my marital relationship, this is unbearable. Who to ask? I need a professional woman to ask her about this sensitive subject, imagine me asking a professional male, no way"* [36] (p. 1049).

Wider family relationships were also deeply affected when one member experienced SMI. For example, a mother reported that her daughter got divorced because of the daughter's illness and that she could no longer fulfill her duties towards her family such as caring for children or husband:

*". . . . Her husband divorce her as he noted her aggressive behavior with children. . . ."* [35]
(p. 525).

**4. Impact of caring on caregiver's mental health.** The majority of caregivers described their caring experience as stressful and overwhelming and were vulnerable to experiencing mental health issues themselves. Most of the caregivers participating in the included studies spoke about their struggles in providing care, and how their experiences contributed to emotional distress and mental illness.

A female spouse spoke about feeling depressed because of her husband who had a mental disorder. She reported:

*"I have husband; he causes me depression and the doctor told me that I have to have medicine to calm me down. I took sleeping pills to be able to sleep, too much that one can bear"* [36]
(p. 1048).

A mother reported considering committing suicide due to burden of providing care to her relative with schizophrenia disorder, which imposed high stress on her:

*"(In tears) I've also gone to the department of mental health. I'm on medication. Otherwise, I feel myself at loose ends. I even considered committing suicide"* [38] (p. 500).

A family caregiver who had been prescribed psychiatric medication described how they felt:

*"Emotionally I'm dead. My life is in a shamble and I'm madly trying to fix it and I'm not getting anywhere, and I keep trying. All the time, I keep trying. I seem to be getting very, very depressed and upset. I don't really want to take more psychiatric drugs you know"* [5] (p. 4).

When caregivers experienced stress and worry about their responsibilities, they become overwhelmed, leading to feeling burned out. Such experiences could lead to hopelessness and mental health problems. A mother from Saudi Arabia experienced excessive stress due to caring which contributed to emotional distress and exhaustion. She exemplified her experience as follows:

*"I was in psychological distress myself, I felt like I was stuck in a whirlpool, a great big whirlpool. . . I cried very hard, it is such a difficult test from God. . . to have your loved ones stricken by this illness (the caregiver was crying)"* [29] (p. 8)

**5. Stigma.** Stigmatizing attitudes towards people with mental illness and their family members were reported in several studies and sources of stigma were manifold. A mother expressed her concern about the stigma of having a family member with mental illness within her community, she stated:

*"Others told us if we had a kid like this we would have died'. I used to go to their houses, but they would not open [the door] although they were home. They believed that my kid might affect their family, and I was despised a lot"* [34] (p. 899).

A family caregiver was disappointed and upset as they experienced some form of stigma from health care professional, they were criticised as "crazy":

*"We came here only because we did not know where to go. . . And what did the doctor tell me? You also act like a lunatic! I felt like I was on fire. . ."* [37] (p. 165).

Misinterpretation about mental illness in the Iranian community led a daughter caregiver to experience stress and encounter stigmatizing behaviours. Therefore, she refused to disclose her father's illness to the public. She stated:

*"I can't tell anyone. They think differently. Their judgment is not good about mental illness. They'd think of him as a lunatic. A dangerous person who must be left alone and never trusted. As if he has a contagious disease. For people, a mentally ill person is like a murderer or retard or even something worse. Unfortunately, their attitude toward this illness is not good. This makes life very difficult for us. We can't deal with it. People think I'm not normal either because I live with a mentally ill father"* [39] (p. 993).

Also, due to negative conceptions in the Jordanian community about mental illness, two female spouses who participated in AlMakhamreh's (2018) [36] study, hid their husbands' illness because of shame and fear of loss of family reputation. From their behaviours, it seems that family caregivers perpetuated this stigma too. They declared:

*"Of course, I don't want to tell anybody about my husband's illness, I have children and I have daughters it is my family's reputation" (p. 1048).*

*"No one will propose to my daughters if they know that their father is suffering from mental health" (p. 1048).*

**6. Seeking support.**   Family caregivers played an important role in supporting their relatives with SMI. However, family caregivers also needed their own sources of support. All participants in the included studies emphasized the potential benefits of having different forms of support from others who could provide empathy and encouragement. This would assist caregivers in their caring role.

*6.1. Social support*. Many family caregivers experienced a lack of support and assistance from family and acquaintances while struggling with their caring role.

One female spouse shared her experience of lacking social support from family and relatives which had a great impact on her emotional well-being:

*"I expect my family and friends to help and support me. However, there is no help. This makes me feel extremely angry, I'm nervous to anything. I easily annoy the other side of the family"* [40] (p. 1511).

Some family caregivers reported positive experiences of social support. A father shared his experience of receiving local support from neighbours. He stated:

*"My neighbours know this, and they donate money and are sympathetic people, saying that they will always support us"* [34] (p.899).

A father also spoke about his positive experience of getting local support from his neighbours who showed empathy and understanding of his family situation:

*"Neighbours, may God bless them, say that if I need a car or if something happens that require emergency, because I do not own a car, they say that I should not feel shy and should ask them for help and that they will readily provide me with the car"* [41] (p. 43).

*6.2. Lack of professional support.* Healthcare professionals can play a potentially critical role in improving family caregivers' experiences. However, all caregivers interviewed in the selected studies experienced a lack of professional support. Not receiving adequate support from healthcare professionals increased the burden on caregivers, as they were unaware about the nature of the disorder, methods of treatment, and ways to manage high-risk behaviours of their loved one with SMI.

A mother was not involved in her son's treatment plan. She did not receive any support from the physician, who did not pay attention to the caregivers' concerns and questions. She declared:

*"I don't know anything about my son's illness, only the doctors said to me that your son had schizophrenia when I asked them more information, they said to me what makes a difference to you that he has what disease, just know he's sick now"* [28] (p. 5).

Family caregivers reported a lack of communication from healthcare professionals. They did not provide appropriate education to family caregivers about their relatives' diagnosis and they felt ignored:

*"They [the psychiatrists] never told us what they think he had. We go in and out of here like ghosts, no one knows who we are, and they never contacted us. . . He comes and goes and we have no idea what is happening here during the admissions. . . We have no one to ask. . . Maybe it is different because it is a psychiatric ward"* [37] (p.166)

Lack of professional support and educational materials resulted in poor understanding of the illness. A family caregiver reported:

*". . . He has been getting treatment from the governmental centre for 10–12 years without any educational support from the centre like teaching us or distributing brochures about the illness. . . .."* [35] (p. 522)

Poor communication by healthcare professionals increased the burden on caregivers. However, family caregivers seemed hopeful about professional support should it arise. One caregiver who regularly visited the mental health service spoke about their positive experience of using healthcare services:

*"This is the first time one care [sic], ask about us as carers, I have been 12 years close to this place I know no one here, I did not want to talk but if we have more of this type of meeting it will be great . . . this will be my shelter"* [36] (p. 1050).

**7. Attempts to cope.** Some family caregivers experienced positive ways of coping with their role by attending group therapies to socialize with other caregivers who may also have had similar experiences of caregiving. A caregiver shared their first-time experience of joining a therapy group and meeting other people who shared a similar situation, which she described as a positive experience:

*"Never met people that have same problem as mine this is the first time; I wish to have more meetings. I feel relaxed already, we are not alone. We learn from each other"* [36] (p. 1050).

Some family caregivers sought information as a positive coping mechanism and at the same time it helps them to understand the illness and raise awareness:

*". . . mainly I was reading about it, the more I learned, the less fearful I became"* [29] (p. 9).

Other family caregivers did not find group therapy a positive way of coping with the situation. One mother expressed that there is no benefit from joining therapy groups, as each member spoke for themselves:

*"Some time ago, someone called me from the hospital and asked me to participate in a group therapy session at the hospital. When I went there, the number of family caregivers was there, and everyone spoke for themselves. I couldn't find why I had been invited there"* [28] (p. 6).

## Discussion

To our knowledge, this is the first review that captures the experiences of family caregivers living in the Middle East caring for relatives with SMI. We conducted this review to comprehensively gain a thorough and detailed overview of what is known about family caregivers' experiences from published qualitative research in the Middle East geographical area to date. Twelve qualitative studies were included in this review. Through meta-synthesis we identified seven overarching themes about the experiences of family caregivers caring for their loved one with SMI. The findings demonstrated that caregivers experienced: 1) difficulties in accepting the diagnosis of their relatives and were often in denial; 2) experienced different types of burdens; 3) faced issues with their family/marital relationships; 4) experienced negative impact on their mental health; 5) stigma; 6) lack of support; and 7) they shared their experiences of attempting to cope with the situation.

Caregivers who were interviewed in the included studies lacked knowledge about their relatives' mental illness, struggled to accept the diagnosis and were often in denial. It is evident that lacking information about the diagnosis was associated with alternative interpretations about mental illness [35,43]. In some Middle Eastern communities, caregivers thought that mental illness manifested due to lack of faith or disconnection from religion [19]. Other communities within the Middle East thought mental illness was attributed to supernatural forces and treatment required spiritual healers [44]. According to Al-Darmaki, Thomas, & Yaaqeib (2016), it is common in most Middle Eastern cultures to believe in supernatural causes of mental disorder such as "jinn (demons) and seher (magic) or evil eye or that it is caused by Allah as a test of faith or punishment for sins" (p. 237) [45]. Whereas in Western cultures, beliefs about biological or psychosocial causes of mental disorders prevail [46]. This shows how family caregivers in the Middle East turned to religious explanations to accept the situation as Allah/God's test. These beliefs strongly influenced how people interpreted mental illness symptoms. As a result, several caregivers turned to traditional or religious healers as the first line of treatment because it is a culturally acceptable and less-stigmatizing way to help people with mental health issues in such countries [36].

In Western cultures, research has shown a significant improvement in people's knowledge, perceptions, and attitudes towards mental illness over time, which has been attributed to increased awareness and social campaigns [47]. However, our study found that mental health stigma in the Middle East was a significant problem for family caregivers as it was often

associated with negative beliefs and stereotypes which affected their lives. Our findings are consistent with a previous systematic review that was conducted on stigma associated with mental illness in the Arab culture, in which the authors found several studies reporting prevailing stigmatizing beliefs and attitudes towards patients with mental illness and their family members [27]. Islam necessitates that vulnerable individuals such as elderly/ill people are protected and looked after, and this would apply to people with mental illness too [48]. Therefore, society has a duty to look after these individuals, and caregivers could be viewed as performing an important role in society, which could lead to more respect and acceptance (and less stigma). However, this was not consistent with the findings of this review and meta-synthesis were stigma and lack of acceptance prevailed.

Most studies included in this review explored caregivers' experiences of burden and the challenges of this caring role. Caregivers reported negative consequences of caring for a relative with SMI without support from others. They were constantly supervising their loved one with SMI, struggling with balancing caring demands, and managing their daily life which involved playing many roles at a time. Other research studies in the Middle Eastern context, have similarly demonstrated that a caring role has a negative impact on caregivers and increases risk of serious health related issues such as posttraumatic stress disorder and depression [21,49,50]. Similar findings have also been found in other countries like China and India, where it has been reported that providing care to people with mental disorders is challenging, leading to severe negative health effects among their carers [2,51,52].

Studies in the West have also shown similar findings to Middle Eastern countries in terms of wellbeing. A European study found that caring for patients with mental disorders confers a substantial burden with negative impact on their health. They also found that the majority of their surveyed caregivers were female family members [53]. A study from the US also found that everyday involvement in most aspects of daily life and the amount of time spent in providing care for people with schizophrenia, increased caregivers' burden, resulting in family caregivers being significantly more likely to experience stress and anxiety [54].

The findings of the current study diverged from findings in Western literature notably around other systems of available support. In Western countries, there is a great available support network for family caregivers such as residential care, support groups, organizations, and respite care centres [55]. However, the findings from this review reveal that family caregivers in Middle Eastern countries were compelled to give care without respite and with few alternatives. The access to professional support and advocacy groups for carers in the Middle East is limited and less well developed compared to other countries, which may explain the high level of carer burden and why they felt unsupported. Caregivers support in the Middle East must be prioritized.

## Strengths and limitations

The strengths of this review are 1) this is the first systematic review and meta-synthesis that focuses on the experiences of family caregivers living in Middle East countries caring for a relative with SMI; 2) the use of qualitative meta-synthesis enabled us to bring qualitative findings together to form an overarching understanding of this topic 3) this review provides a broad understanding about the experiences of family caregivers of people with SMI in the Middle East; 4) two authors independently assessed the titles and abstracts and full texts to ensure that we included the rights studies; and 5) two authors independently assessed the quality of all included studies.

The limitation of the review are 1) due to the current dearth of research in this area, only a small number of studies were identified and included in the review, 2) limited diversity of the

included studies (most of the studies were based in Iran which does not represent caregivers' experiences in other geographical areas in the Middle East), 3) each country had different beliefs and interpretations about mental illness, therefore, the experiences of family caregivers in all included studies cannot be generalised to all Middle East countries. For instance, the understanding and beliefs about mental disorders, the practices of caregiving, and the available support services may not be the same in counties such as Qatar as in Iran. This is due to many cultural differences, varying religious practices between the countries, the practice of mental health, and that the services provided for the public is different, 4) we did not get access to the original data transcripts, therefore, the ability to identifying cross-cutting themes is restricted to what was already reported in the published studies, and 5) the search was limited to English and Arabic languages, there might be studies related to this topic published in other languages such as Farsi or Turkish.

## Implication for future research and practice

The findings of the current study point to a need to urgently reduce the burden and challenges of family caregivers of people with SMI in the Middle East. Since studies consistently demonstrated that family caregivers experience great emotional distress and lack of access to social/healthcare support needed, they are arguably at high risk of psychological ill-health and poor quality of life.

This study highlights significant implications for healthcare professionals. There is a need for healthcare professionals to focus on not only people with SMI, but to extend their care to include family member who provide support to people with SMI at home. Improving family caregiver outcomes requires theoretical and practical understanding of the caring role to reduce burden. Although family caregivers remain the greatest resource of support for their relatives with SMI, they very much merit the attention of healthcare professionals in their own right.

Future research should seek to qualitatively explore experiences of family caregivers in other Middle Eastern countries, not included in this review, where there are high rates of people diagnosed with SMI. To further enrich our understanding of family caregivers' needs and experiences, we recommend conducting further in-depth qualitative research. Further quantitative research is also needed to establish the extent and prevalence of these problems amongst and between caregivers within, and between, different Middle Eastern countries, and to provide a benchmark against which change can be measured.

## Conclusion

The findings of this systematic review and qualitative meta-synthesis highlight the challenges and issues that affect the caregivers of people with SMI in the Middle East. The review identifies education and support needs that could reduce the burden on caregivers and improve their quality of life. Understanding the experiences of family caregivers could better inform healthcare professionals to provide more attention to their needs.

## Supporting information

**S1 File. PRISMA checklist.**
(DOCX)

**S2 File. Full list of search terms.** Showing full list of search terms that were applied in all databases.
(PDF)

## Author Contributions

**Formal analysis:** Aisha Hamed Alyafei.

**Methodology:** Aisha Hamed Alyafei, Taghrid Alqunaibet, Hassan Mansour.

**Supervision:** Afia Ali, Jo Billings.

**Writing – original draft:** Aisha Hamed Alyafei.

**Writing – review & editing:** Aisha Hamed Alyafei, Taghrid Alqunaibet, Hassan Mansour, Afia Ali, Jo Billings.

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
