## [Decision Letter · Decision Letter 0]

12 Apr 2021

PONE-D-21-05863

The experiences of family caregivers of people with severe mental illness in the Middle East: A systematic review and meta-synthesis

PLOS ONE

Dear Dr. Alyafei,

Thank you for submitting your manuscript to PLOS ONE. After careful consideration, we feel that it has merit but does not fully meet PLOS ONE’s publication criteria as it currently stands. Therefore, we invite you to submit a revised version of the manuscript that addresses the points raised during the review process.

We look forward to receiving your revised manuscript.

Kind regards,

Perla Werner

Academic Editor

PLOS ONE

Journal Requirements:

2. Please note that in order to use the direct billing option the corresponding author must be affiliated with the chosen institute. Please either amend your manuscript to change the affiliation or corresponding author, or email us at plosone@plos.org with a request to remove this option.

3. Please include captions for all your Supporting Information files at the end of your manuscript, and update any in-text citations to match accordingly. Please see our Supporting Information guidelines for more information: http://journals.plos.org/plosone/s/supporting-information.

Reviewers' comments:

Reviewer's Responses to Questions

**Comments to the Author**

1. Is the manuscript technically sound, and do the data support the conclusions?

Reviewer #1: Partly

Reviewer #2: Yes

2. Has the statistical analysis been performed appropriately and rigorously? 

Reviewer #1: N/A

Reviewer #2: N/A

3. Have the authors made all data underlying the findings in their manuscript fully available?

Reviewer #1: Yes

Reviewer #2: Yes

4. Is the manuscript presented in an intelligible fashion and written in standard English?

Reviewer #1: No

Reviewer #2: Yes

5. Review Comments to the Author

Reviewer #1: Thank you for the opportunity to review this systematic review and meta-synthesis of the family caregivers' experiences of people with SMI in the Middle East. This is an important topic, and the authors should be commended for conducting a study among family caregivers in this specific geographic area. There are some fundamental issues with the manuscript in its current form, however, that I think ought to be addressed before it would be suitable for publication, particularly regarding the introduction, results and discussion sections. I outline these below, point by point.

Title

1. I recommend clarifying in the title that the systematic review is of qualitative data. the authors may consider the title "The experiences of family caregivers of people with severe mental illness in the Middle East: A systematic review and meta-synthesis of qualitative data".

Abstract

1. In the conclusions, what do you mean by "Family caregivers experienced distress and burden which had a significant impact on their psychological well-being"? I think this sentence needs to be rewritten, it has not been directly examined and it is problematic to talk about a significant impact in this manuscript.

Introduction

1. When authors present data or refer to conclusions from previous studies they should include citations. For example, see lines 54-56, page 4 (suggestion for citation: World Health Organization (2019). Mental disorders. Retrieved from:

https://www.who.int/news-room/fact-sheets/detail/mental-disorders). Also see, lines 61 and 66, page 4, and line 90, page 5 (suggestion for citation: Abo-Rass, F., et al., (2020). Depression Illness Representations Among Arabs in Israel: A Qualitative Study Comparing Younger and Older Adults. Journal of Cross-Cultural Gerontology, 35(4), 353-366.‏ and Abu-Kaf, S., & Braun-Lewensohn, O. (2015). Paths to depression among two different cultural contexts: Comparing Bedouin Arab and Jewish students. Journal of Cross-Cultural Psychology, 46, 612-630).

2. Paragraph 2, page 4 - The consequences of caring for a family member with MSI should be expanded.

3. The ideas of paragraphs 2 and 3 on page 4 (not the content) should be connected so the general idea will be clearer. You might start paragraph 3 with something as: unlike the western countries, in the Middle East, very little research exists regarding ... that is despite ...

4. In lines 83-84, page 5, it is mentioned "Few studies have explored in depth the

experiences of caregivers in this region using qualitative methodologies", these studies should be presented, also there's findings and conclusions. (suggestions for this, see: Abojabel, H., & Werner, P. (2019). Exploring family stigma among caregivers of persons with Alzheimer's disease: The experiences of Israeli-Arab caregivers. Dementia, 18(1), 391-408).

5. Lines 92-93, page 5, "Whilst Middle Eastern countries can be different from the West there is also great variation within different Middle Eastern countries". Which variation do you mean? The authors should clarify this sentence to avoid confusion

6. Literature should be added regarding the status/importance of the family among individuals in the Middle East, and in Arab and Muslim culture.

7. Lines 71-75, page 4; as there are several definitions for countries included in the Middle East, please clarify the source of the Middle East definition and add a citation.

Method

1. I suggest merging the sections "design" and "search strategy". In addition, the rationale of the systematic methodology should appear in the introduction. Please make sure there is no repetition.

2. The section "selection criteria" should be focused and shortened. For example: no need for criteria No.4 in the excluded criteria.

3. In the section "selection criteria", inclusion criteria No.7 is not clear. Does it include major depression, dementia and Alzheimer? I am familiar with other studies that have qualitatively examined the experience of family caregivers for individuals with dementia in the Middle East. This criterion must be redefined.

Results

1. Regarding the "Themes" section; the section is very long; it should be reduced to fewer themes. For example, it is not clear how the sub-themes lack of patient's insight into the illness is related? However, you may combine it with the sub-theme lack of knowledge. Similar to this, the sub-themes "Restrictions", " Sole Responsibility" and " Self-Sacrifice" might be combined. I recommend rearranging and rewriting the themes. I also suggest reducing the number of quotes per theme and choosing only the most appropriate one.

2. In the "themes" section, as table 3 shows the "Overarching themes emerging from meta-synthesis", no need for figure 2.

Discussion

1. The authors could better explain and integrate the findings with existing literature.

Strengths and limitations

1. In order to avoid repetition, it is recommended to focus only on the limitations of the review. the strengths and limitations of the included studies have already been presented throughout the article.

Implication for future research and practice

1. In line 767, page 34, what different qualitative research do you mean? Please detail it.

The language of the current manuscript is unclear, making it difficult to follow. I advise the authors to improve the flow and readability of the text.

Reviewer #2: This is a rigorous and well-written review of an area of literature that has not been previously addressed in a review article.

It could be made more clear in the title that the review only focuses on qualitative studies, as this wasn’t evident to me until I started reading the body of the article.

An additional question I have is whether the focus on the Middle East is purely geographic, or if the focus is specifically on Muslim families within the Middle East. There have been a couple of qualitative studies conducted in Israel that included family members (Amsalem, D., Hasson-Ohayon, I., Gothelf, D., & Roe, D. (2018). Subtle ways of stigmatization among professionals: The subjective experience of consumers and their family members. Psychiatric Rehabilitation Journal, 41(3), 163-168. doi:10.1037/prj0000310; Amsalem, D., Hasson-Ohayon, I., Gothelf, D., & Roe, D. (2018). How do patients with schizophrenia and their families learn about the diagnosis? Psychiatry (New York), 81(3), 283-287. doi:10.1080/00332747.2018.1443676) that may have been excluded as they did not focus on Muslim families, but this should at least be clarified, as they otherwise appear to fit within the inclusion criteria.

I understand that the article followed rigorous strategies with regard to the meta-synthesis approach, but I think that some limitations of it should be acknowledged. The authors did not, as I understand it, get access to the original data transcripts, so their ability to identifying cross-cutting themes is restricted to what was already reported in the published studies. This may mean that the identification of new themes can’t truly be separated from what was identified by the original study authors, and should be noted as a limitation, as one of the purposes of the study was to identify “cross-cutting” themes.

Something else that I wonder if could be brought in in the discussion section is the way in which expectations for the care of the “incompetent,” which has been identified as a factor that might increase social acceptance and decrease stigma within Muslim communities (Youssef, H. A., & Youssef, F. A. (1996). Evidence for the existence of schizophrenia in medieval Islamic society. History of Psychiatry, 7, 55– 62.) might also increase burden among family members.

6. PLOS authors have the option to publish the peer review history of their article (what does this mean?). If published, this will include your full peer review and any attached files.

Reviewer #1: No

Reviewer #2: No

---

## [Author Response · Author response to Decision Letter 0]

10 Jun 2021

Please see response letter which has been uploaded.

---

## [Decision Letter · Decision Letter 1]

25 Jun 2021

The experiences of family caregivers of people with severe mental illness in the Middle East: A systematic review and meta-synthesis of qualitative data

PONE-D-21-05863R1

Dear Dr. Alyafei,

We’re pleased to inform you that your manuscript has been judged scientifically suitable for publication and will be formally accepted for publication once it meets all outstanding technical requirements.

Kind regards,

Perla Werner

Academic Editor

PLOS ONE

Additional Editor Comments (optional):

Reviewers' comments:

Reviewer's Responses to Questions

**Comments to the Author**

1. If the authors have adequately addressed your comments raised in a previous round of review and you feel that this manuscript is now acceptable for publication, you may indicate that here to bypass the “Comments to the Author” section, enter your conflict of interest statement in the “Confidential to Editor” section, and submit your "Accept" recommendation.

Reviewer #1: All comments have been addressed

2. Is the manuscript technically sound, and do the data support the conclusions?

Reviewer #1: Yes

3. Has the statistical analysis been performed appropriately and rigorously? 

Reviewer #1: Yes

4. Have the authors made all data underlying the findings in their manuscript fully available?

Reviewer #1: Yes

5. Is the manuscript presented in an intelligible fashion and written in standard English?

Reviewer #1: Yes

6. Review Comments to the Author

Reviewer #1: the authors have addressed my comments in the previous review and i suggest this manuscript is now acceptable for publication

7. PLOS authors have the option to publish the peer review history of their article (what does this mean?). If published, this will include your full peer review and any attached files.

Reviewer #1: No

---

## [Editor Report · Acceptance letter]

30 Jun 2021

PONE-D-21-05863R1 

The experiences of family caregivers of people with severe mental illness in the Middle East: A systematic review and meta-synthesis of qualitative data 

Dear Dr. Alyafei:

I'm pleased to inform you that your manuscript has been deemed suitable for publication in PLOS ONE. Congratulations! Your manuscript is now with our production department. 

Kind regards, 

on behalf of

Professor Perla Werner 

Academic Editor

PLOS ONE